# GAAGs, COMP, and YKL-40 as Potential Markers of Cartilage Turnover in Blood of Children with Juvenile Idiopathic Arthritis Treated with Etanercept—Relationship with ADAMTS4, ADAMTS5, and PDGF-BB

**DOI:** 10.3390/jcm11175069

**Published:** 2022-08-29

**Authors:** Klaudia Dąbkowska, Magdalena Wojdas, Kornelia Kuźnik-Trocha, Grzegorz Wisowski, Anna Gruenpeter, Katarzyna Komosińska-Vassev, Krystyna Olczyk, Katarzyna Winsz-Szczotka

**Affiliations:** 1Department of Clinical Chemistry and Laboratory Diagnostics, Faculty of Pharmaceutical Sciences in Sosnowiec, Medical University of Silesia, ul. Jedności 8, 41-200 Sosnowiec, Poland; 2Department of Rheumatology, The John Paul II Pediatric Center in Sosnowiec, ul. G. Zapolskiej 3, 41-218 Sosnowiec, Poland

**Keywords:** juvenile idiopathic arthritis, etanercept, extracellular matrix turnover markers, galactosaminoglycans, oligomeric cartilage matrix protein, human cartilage glycoprotein 39

## Abstract

We quantified galactosaminoglycans (GAAGs), oligomeric cartilage matrix protein (COMP), and human cartilage glycoprotein 39 (YKL-40) in blood obtained from juvenile idiopathic arthritis (JIA) before and during 2-year treatment with etanercept (ETA), as potential biomarkers of cartilage extracellular matrix (ECM) dysfunction and indicators of efficacy of biologic therapy. We also evaluated the relationship of the mentioned markers with the factors that regulate their metabolism, disintegrin and thrombospondin motif metalloproteinases 4 (ADAMTS4), ADAMTS5, and platelet-derived growth factor BB (PDGF-BB). Methods: We studied 38 children diagnosed with JIA and 45 healthy children. We quantified GAAGs by assessing the concentration of unsaturated disaccharide units formed by digestion of isolated glycosaminoglycans with chondroitinase ABC, while COMP, YKL-40, and PDGF-BB were quantified using immunoenzymatic methods. Results: Compared to the control group, GAAGs and COMP levels were significantly lower, while YKL-40 levels were higher in the blood of patients with aggressive JIA, qualified for ETA treatment. ETA therapy leading to clinical improvement simultaneously promoted normalization of COMP and YKL-40 levels, but not GAAGs. After 24 months of taking ETA, glycan levels were still significantly lower, relative to controls. GAAGs, COMP, and YKL-40 levels were significantly influenced by ADAMTS4, ADAMTS5, and PDGF-BB levels both before and during ETA treatment. Conclusions: The dynamics of changes in marker concentrations during treatment seem to indicate that measurement of COMP and YKL-40 levels can be used to assess the chondroprotective biological efficacy of therapy. In contrast, changes in GAAGs concentrations reflect systemic extracellular matrix transformations in the course of JIA.

## 1. Introduction

Juvenile idiopathic arthritis (JIA), being the most frequently diagnosed heterogeneous group of rheumatic diseases of developmental age, is the main cause of disability in children. Degradation of the osteoarticular system in the course of arthropathy is connected with hypersecretion of proinflammatory cytokines, including tumor necrosis factor alpha (TNF-α), interleukin (IL) 1β, or IL-6 [1,2]. These molecules, through numerous mechanisms including regulation of disintegrin and thrombospondin motif metalloproteinases (ADAMTS) activity or growth factors, including platelet-derived growth factor BB (PDGF-BB), promote remodeling of the three-dimensional structure of cartilage extracellular matrix (ECM). As a result of pathological remodeling of the ECM, unique biophysical properties of cartilage, such as elasticity, viscosity, and resistance to stress, are altered. These biomechanical properties are related to the organization of matrix macromolecules. The latter include structural proteins, mainly collagen type II, non-collagenous glycoproteins, including fibronectin, laminin, or entactin, as well as proteoglycans (PGs), including aggrecan or decorin [3,4,5]. The maintenance of normal, age-dependent ECM structure is also determined by the effects of oligomeric cartilage matrix protein (COMP) as well as 40 kDa human cartilage glycoprotein 39 (YKL-40). The disturbances of the physiological balance between decomposition and restoration of the mentioned components of ECM leads to progressive loss of osteoarticular system. It has been suggested that the degradation of cartilage proteoglycans, co-formed by galactosaminoglycan chains (GAAGs), represents an early critical incident in the articular cartilage injuries, and at the same time is likely to precede irreversible changes of the cartilage collagen weave [6,7,8].

Therefore, early detection of structural abnormalities of the cartilage is very important. This will allow clinicians to quickly initiate accurate therapies, which is essential for the course of JIA. Delayed diagnosis of pathological changes in joints, due to the lack of characteristic diagnostic biomarkers, determined in available biological material, may result in the perpetuation of dysfunction in the musculoskeletal system [9]. The biomarkers of cartilage transformations may be matrix components as well as their derivatives, which are released during metabolic transformations into biological fluids. In our previous studies, we evaluated specific markers of cartilage ECM remodeling, including procollagen II C-terminal propeptide (PIICP), C-telopeptide of type II collagen (CTXII) [10], hyaluronic acid, keratan sulfates, hyaluronan and proteoglycan link protein 1 [11], or metalloproteinases [12]. Considering the above, this study was designed to evaluate the yet unknown dynamics of changes in the levels of GAAGs, COMP, and YKL-40-as potential biomarkers of dysfunctions of the osteoarticular system and the efficacy of biological therapy in the blood of JIA patients before and during 2 years of treatment with TNF-α inhibitors. Since the metabolism of the evaluated matrix components is related to the activity of proteolytic enzymes of the ADAMTS group, i.e., ADAMTS4, ADAMTS5, as well as the factor stimulating their anabolic metabolism, i.e., PDGF-BB, we decided to assess their interrelationships [3,13]. The plasma ADAMTS4 and ADAMTS5 concentrations in JIA patients before and after 24 months of ETA treatment were previously published by our research team [11].

## 2. Materials and Methods

### 2.1. Patients and Samples

In the current study, 38 Polish Caucasian patients, aged 4 to 12 years, of both sexes, with a diagnosis of JIA according to ILAR (International League of Associations for Rheumatology) criteria [13], were selected from the Rheumatology Clinic of the John Paul II Pediatrics Center in Sosnowiec, Poland, and were qualified for biological treatment with ETA.

Disease activity in children was assessed on the basis of variables including both the physician’s total assessment of disease activity, measured on a 10 cm visual analog scale (VAS scale, where 0 = inactive state, 10 = maximum activity), the child’s or caregiver’s assessment of well-being (VAS scale), and the number of joints with signs of active inflammation (swelling or limited mobility and increased warmth, pain or tenderness). In addition, laboratory results confirmed the accuracy of the diagnosis (Table 1).

Patients with musculoskeletal trauma or surgery within the past three years, patients with autoimmune diseases, patients with metabolic diseases such as diabetes, cancer, kidney disease, liver disease, and chronic infections, and patients with mental illness or other conditions that impede physician–patient collaboration were excluded from the study.

For JIA patients with arthropathy, sulfasalazine (SSA, Sulfasalazine EN) used at a dose of 30 mg per kilogram of body weight, prednisone (Encorton, EC) at a dose of 1 mg per kilogram of body weight, and methotrexate (MTX) at a dose of ≤15 mg per square meter of body surface area, at one dose per week were used. American College of Rheumatology (ACR) patient improvement criteria were used [14]. In this work, we evaluated plasma ECM biomarkers of cartilage in patients whose condition was not improved after three months of treatment with these drugs. The above results were the basis for implementing ETA therapy in these patients (all JIA patients participated in the National Health Fund Therapeutic Programs with TNF blockers—that is, B.33). Etanercept was administered by subcutaneous injection twice a week, with breaks every 3 to 4 days, at a dose of 0.4 mg/kg body weight or 0.8 mg/kg body weight once a week.

Patients qualified for treatment with TNF-α inhibitor exhibited features of polyarticular form of JIA or oligoarticular JIA (prolonged and persistent) with poor prognosis factors. The exclusion criteria were other forms of JIA and other chronic and autoimmune diseases, previous treatment with biologic drugs, and discontinuation of biologic therapy during the study period.

In all JIA patients, circulating blood biomarkers of ECM changes were assessed both before the start of ETA treatment therapy (T0) and in the same patients after the 3rd (T3), 6th (T6), 12th (T12), 18th (T18), and 24th (T24) month of biological therapy.

Control samples were obtained from 45 healthy individuals whose age and gender matched the JIA patients. Children in the control group did not suffer from any diseases requiring hospitalization and had not undergone surgery within the past year. The clinical data of the healthy subjects and JIA patients included in the study are shown in Table 1.

**Table 1 jcm-11-05069-t001:** Clinical data of healthy patients and patients on biological therapy.

Parameter	Control Subjects (n = 45)	Patients of JIA (n = 38)					
		Before ETA Treatment T0 (n = 38)	Time after Starting Etanercept Therapy				
			3 Months T3	6 Months T6	12 Months T12	18 Months T18	24 Months T24
Age (years)	8.01 ± 2.59	6.82 ± 2.04	7.04 ± 2.08	7.31 ± 2.07	7.82 ± 2.03	8.37 ± 2.71	8.84 ± 2.07
Sex (F/M)	21/9	25/10	25/10	25/10	25/10	25/10	25/10
JADAS-27	-	41.50 (36.50–49.50)	17.50 (15.50–21.50)	9.50 (8.00–13.50)	2.50 (1.00–4.00)	1.00 (1.00–1.50)	0.50 (0.00–1.00) ^b^
Treatment drugs	-	MTX, EC, SSA	ETA, MTX, EC, SSA	ETA, MTX	ETA, MTX	ETA, MTX	ETA, MTX
WBC (10^3^/μL)	5.23 ± 2.15	9.88 ± 3.70 ^a^	7.07 ± 2.63	6.96 ± 2.85	6.75 ± 1.52	6.52 ± 1.62	6.28 ± 2.16 ^b^
RBC (10^6^/μL)	4.85 ± 0.33	3.87 ± 0.58 ^a^	4.51 ± 0.72	4.50 ± 0.82	4.48 ± 0.32	4.60 ± 0.39	4.86 ± 0.64
Hb (g/dL)	13.84 ± 1.81	11.35 ± 2.52 ^a^	11.99 ± 1.95	12.61 ± 4.40	13.50 ± 1.81	13.01 ± 1.46	13.80 ± 1.25 ^b^
PLT (10^3^/μL)	293.20 ± 71.56	348.95 ± 55.04	362.41 ± 53.88	318.95 ± 77.10	327.74 ± 84.96	312.05 ± 78.96	336.15 ± 50.50
GPT (U/L)	19.65 ± 7.98	23.96 ± 11.02	22.25 ± 7.41	17.56 ± 11.00	21.08 ± 8.30	24.45 ± 7.61	26.00 ± 10.08 ^a^
GOT (U/L)	25.68 ± 9.02	26.99 ± 10.98	26.70 ± 7.82	22.28 ± 7.41	22.19 ± 10.47	23.22 ± 10.87	25.92 ± 12.20
Cr (mg/dL)	0.69 ± 0.42	0.68 ± 0.55	0.70 ± 0.72	0.65 ± 0.23	0.70 ± 0.25	0.83 ± 0.25	0.96 ± 0.50 ^b^
ESR (mm/h)	6.99 ± 2.21	42.85 ± 13.27 ^a^	29.41 ± 13.05	12.04 ± 7.84	8.95 ± 2.49	9.65 ± 6.60	8.87 ± 1.23 ^b^
CRP (mg/L)	0.67 (0.36–1.00)	23.83 (18.5–33.79) ^a^	13.98 (11.69–16.12)	0.79 (0.38–5.16)	0.74 (0.32–2.6)	0.45 (0.2–1.2)	0.43 (0.23–1.61) ^b^
ADAMTS4 * (ng/mL)	21.02 ± 9.01 (n = 45)	38.20 ± 10.94 ^a^ (n = 54)	32.16 ± 9.87 (n = 54)	29.58 ± 8.98 ^b^ (n = 54)	34.36 ± 9.55 (n = 54)	36.68 ± 12.98 (n = 54)	26.96 ± 10.15 ^b,c^ (n = 54)
ADAMTS5 * (ng/mL)	28.66 ± 6.66 (n = 45)	38.09 ± 15.60 ^a^ (n = 54)	30.08 ± 12.25 (n = 54)	23.89 ± 7.95 ^b^ (n = 54)	23.35 ± 8.50 ^b^ (n = 54)	22.84 ± 6.09 ^b^ (n = 54)	24.05 ± 5.66 ^b^ (n = 54)

Results are expressed as mean ± SD or medians (quartile 1–quartile 3); ETA, etanercept; F/M, female/male; JADAS-27, Juvenile Arthritis Disease Activity Score-27; JIA, Juvenile idiopathic arthritis; MTX, methotrexate; EC, encorton; SSA, sulfasalazin; WBC, white blood cell; RBC, red blood cell; Hb, hemoglobin; PLT, platelet; GPT, glutamic pyruvic transferase; GOT, glutamic oxaloacetic transaminase; Cr, creatinine; ESR, erythrocyte sedimentation rate; CRP, C-reactive protein; ADAMTS4, disintegrin and metalloproteinase with thrombospondin motifs 4; ADAMTS5, disintegrin and metalloproteinase with thrombospondin motifs 5; ^a^
*p* < 0.05 compared to control group; ^b^
*p* < 0.05 compared to T0 group; ^c^
*p* < 0.05 compared to T18 group; * results have been published in previous report [11].

Detailed characteristics of the study group and the healthy subjects were included in our earlier work [11]. The present paper is based on analyses conducted on the same group of JIA patients and healthy children, as it is part of a whole series of studies. All subjects gave their informed consent to the inclusion prior to participating in the study.

The investigation was performed in accordance with the principles of the Helsinki Declaration, as well as being approved by the the Local Bioethics Committee of the Medical University of Silesia in Katowice (number of consent: KNW/0022/KB/168/18).

### 2.2. Isolation of Plasma Glycosaminoglycans

The total pool of circulating blood glycosaminoglycans, including galactosaminoglycans, consists of native blood glycosaminoglycans, glycan chains derived from the breakdown of tissue PGs and blood PGs, and free hyaluronic acid chains. Isolation of glycosaminoglycans from blood plasma was performed according to the method of Volpi et al. [14], modified by Olczyk et al. [15], as previously described [16].

### 2.3. Determination of the Concentration of Hexuronic Acids

A measure of the amount of glycosaminoglycans is the concentration of hexuronic acids, which contribute to the disaccharide units of glycan chains. The concentration of hexuronic acids was determined by the carbazole method of Volpi et al. [17], Filisetti-Cozzi and Carpita [18], and van den Hoogen et al. [19].

A detailed procedure for the determination of the concentration of hexuronic acids is presented in our earlier work [12].

### 2.4. Quantitative Evaluation of Galactosaminoglycans

Galactosaminoglycans were quantified by evaluating the concentration of unsaturated disaccharide units formed by digestion of isolated glycosaminoglycans with chondroitinase ABC—an enzyme that specifically depolymerizes galactosaminoglycans. Chondroitinase ABC [E.C.4.2.2.4] is a bacterial lyase that breaks the glycosidic bond of type β (1→4) between the N-acetylgalactosamine (GalNAc) and hexuronic acid (HexA) residues within galactosaminoglycans. The products of the action of the above-mentioned bacterial lyase on GAAGs chains are unsaturated disaccharides, of the formula ∆HexA-GalNAc, and resistant to the action of chondroitinase ABC, six-sugar fragments of chondroitin, and dermatan sulfate chains. Degradation of GAAGs chains at 4 μg hexuronic acids with chondroitinase ABC was carried out in Tris-HCl buffer, pH 8.0, for 30 min at 37 °C. An enzyme with an activity of 0.02 IU was used to completely depolymerize the fraction of GAAGs susceptible to digestion by the lase in question. Both before enzyme addition and after digestion the absorbance of the test samples and standard samples of dermatan sulfate solutions were measured at λ = 232 nm using a microplate reader (Infinite M200, Tecan, Grödig, Austria). The concentration of individual unsaturated disaccharides, derived from GAAGs digestion of plasma from healthy subjects and from subjects with JIA, was determined from a calibration curve made for DS standard solutions with concentrations of 20, 15, 10, 7.5, 5, 2.5, or 1.25 μg/mL. The intra-assay variability was less than 5%.

### 2.5. The Assay of the Concentration of COMP, YKL-40, and PDGF-BB

Levels of COMP, YKL-40, and PDGF-BB were measured in duplicate using blinded plasma samples with a code. Since the surveys were conducted within one day, it was insignificant variability between samples. The markers evaluated were determined by enzyme-linked immunosorbent assays (ELISA) according to the recommendation provided by the assay manufacturer. The tests used were intended solely for scientific research. The determination of plasma COMP concentration was performed with the Human Cartilage Oligomeric Matrix Protein Test Kit by BioVendor Research and Diagnostic Product (Brno, Czech Republic), with a minimum detection of 0.4 ng/mL, whereas YKL-40 plasma concentration was measured with the YKL-40 EIA Test Kit by MicroVue (San Diego, CA, USA), with a minimum detection of 5.4 ng/mL. In turn, the concentration of plasma PDGF-BB was determined with ELISA Kits by Cloud-Clone Corp. (Houston, TX, USA), with a minimum detection of 28.3 pg/mL. The intra-assay variability, for all tested markers, was less than 6%.

### 2.6. Statistical Analysis

The obtained results were statistically analyzed using the computer program STATISTICA (version 13.3 by TIBCO StatSoft, Krakow, Poland). The analysis included normality of the distribution for a given group checked using the Shapiro–Wilk test, Levene’s test for equality of variance, descriptive characteristics for normally distributed characteristics in the form of arithmetic mean as a measure of location and standard deviation as a measure of variability, and testing the significance of differences in the mean values of a given trait for control and tested groups for traits with normal distribution by applying the Student’s *t*-test for independent samples; for skewed distribution Kruskal–Wallis test was used. Significance testing of the differences in the mean values of a given trait for subjects in each study group (from each treatment time) was performed for traits with a normal distribution by using analysis of variance (ANOVA) with repeated measures. As a post-hoc test, Tukey’s multiple comparisons test was used. The strengths of correlations between traits were determined using Pearson’s correlation coefficient, which was modified using Bonferroni’s multivariate adjustment. *p*-Values of less than 0.05 were considered significant.

## 3. Results

The levels of GAAGs, COMP, YKL-40, and PDGF-BB were assessed only in JIA patients who achieved remission during ETA treatment.

### 3.1. The Plasma Levels of GAAGs, COMP, YKL-40, and PDGF-BB in Healthy Children and JIA Patients

As a result of our study, we found that in the course of JIA, the metabolism of heteropolysaccharide components of the connective tissue extracellular matrix was disturbed, manifested by quantitative changes in plasma concentrations of GAAGs, COMP, and YKL-40 in patients (Table 2). In the blood of children with untreated biologic JIA (T0), there was a statistically significant decrease in the concentrations of GAAGs and COMP (*p* < 0.001), by 51% and 22%, respectively, and a statistically significant increase in the concentration of YKL-40 by 97% (*p* < 0.001) compared to the concentrations of the evaluated matrix components in the blood of healthy children. Moreover, it was shown that the two-year application of anti-cytokine therapy, leading to clinical improvement in patients, simultaneously contributed to normalization of the concentration of COMP and YKL-40 only. The concentration of GAAGs in the blood of children after the two-year period of ETA treatment was still significantly lower (*p* = 0.004) in relation to the plasma glycans concentration in the control group.

To evaluate factors regulating ECM metabolism, PDGF-BB levels were determined in blood samples from control subjects and JIA patients (Table 2). Quantitative analysis of the assessed marker revealed a statistically significant (*p* < 0.001, 51%) increase in plasma PDGF-BB levels in children with JIA qualified for etanercept treatment (T0), as compared to the control group. Moreover, it was shown that the two-year application of anti-cytokine therapy in patients, contributing to clinical improvement, simultaneously leads to normalization of the concentration of the assessed marker (*p* = 0.96).

### 3.2. Changes in Plasma Levels of GAAGs, COMP, YKL-40, and PDGF-BB in Patients with JIA during ETA Treament

The levels of the assessed markers in the patients’ blood samples collected before the administration of the drug (T0) and after 3 (T3), 6 (T6), 12 (T12), 18 (T18) and 24 (T24) months of applying ETA are presented in Table 2 and visualized in Figure 1a–d.

Different trends of changes in the concentrations of GAAGs, COMP, and YKL-40 in the blood of patients during the two-year period of biological treatment were demonstrated. In the case of GAAGs, two trends of changes in plasma concentrations of these compounds were observed during etanercept therapy, i.e., the first, increasing trend was observed until the 6th month of therapy (T6), whereas the continuation of ETA in patients with arthropathy led to a gradual decrease in plasma concentrations of circulating glycans until the 24th month of therapy (Figure 1a). The concentration of GAAGs in the blood of patients with active arthropathy qualified for biological treatment (T0) was significantly lower compared to the concentration in the blood of patients in subsequent months of treatment, i.e., T3 (*p* = 0.000020), T6 (*p* = 0.000020), T12 (*p* = 0.000020), T18 (*p* = 0.000020), and T24 (*p* = 0.000020). Moreover, it was shown that the highest concentration of GAAGs found in the plasma of patients in the sixth month of anticytokine therapy (T6) was significantly different from the concentrations in patients from the following groups: T12 (*p* = 0.000160), T18 (*p* = 0.000020), and T24 (*p* = 0.000020). There were also significant differences in GAAGs levels between patients in groups T3 and T12 (*p* = 0.017746), T3 and T18 (*p* = 0.000020), T3 and T24 (*p* = 0.000020), T12 and T18 (*p* = 0.007706).

The less intense, as compared to GAAGs, changes of plasma COMP concentrations were demonstrated being manifested by a gradual increase of the assessed protein in the blood of biologically treated patients (Figure 1b). The statistically significant differences in plasma COMP concentrations between the group of children qualified for biological therapy (T0) and patients from groups T3 (*p* = 0.000169), T6 (*p* = 0.000076), T12 (*p* = 0.000021), T18 (*p* = 0.000026), and T24 (*p* = 0.000192) were found.

In contrast to GAAGs and COMP, the dynamics of changes in the concentration of YKL-40 was characterized by a decreasing tendency with the duration of biological therapy (Figure 1c). Thus, differences were found between the concentration of the assessed glycoprotein in blood of untreated patients with JIA (T0) and concentrations characterizing the same patients from the following groups: T3 (*p* = 0.000020), T6 (*p* = 0.000020), T12 (*p* = 0.000020), T18 (*p* = 0.000020), and T24 (*p* = 0.000020). There were also significant differences in YKL-40 levels between patients in the T3 and T24 groups (*p* = 0.001356).

The trends of changes in plasma PDGF-BB concentrations in children with JIA during the two-year etanercept therapy are similar to those of YKL-40 (Figure 1d). The highest value of PDGF-BB concentration in blood obtained from untreated patients qualified for ETA treatment (T0) was significantly different from concentrations in the same patients in subsequent months of therapy: T6 (*p* = 0.000954), T12 (*p* = 0.000023), T18 (*p* = 0.000102), and T24 (*p* = 0.000021). There were also significant differences in PDGF-BB levels between patients in the T3 and T24 groups (*p* = 0.025617).

**Figure 1 jcm-11-05069-f001:**
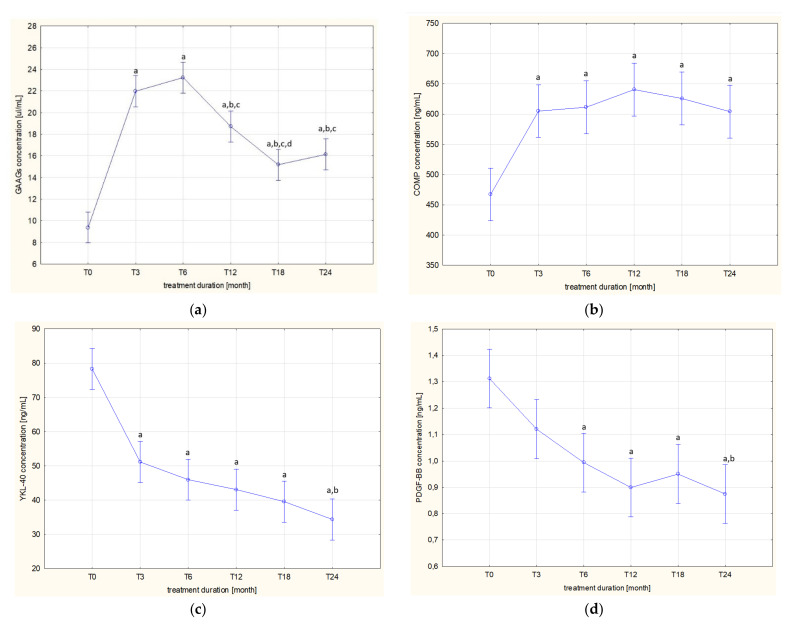
Dynamics of changes in plasma levels of GAAGs (**a**), COMP (**b**), YKL-40 (**c**), and PDGF-BB (**d**) in patients with JIA before and during etanercept therapy; ^a^
*p* < 0.05 compared to T0 group; ^b^
*p* < 0.05 compared to T3 group; ^c^
*p* < 0.05 compared to T6 group; ^d^
*p* < 0.05 compared to T12 group.

### 3.3. Correlation Analysis between Plasma GAAGs, COMP, and YKL-40, and ADAMTS4, ADAMTS5, and PDGF-BB Levels in JIA Patients

In order to achieve the primary aim of the study, we also assessed the association of GAAGs, COMP, and YKL-40 plasma concentrations in children with JIA, both before and during the two-year ETA therapy, with the concentrations of compounds affecting their metabolism, i.e., ADAMTS4, ADAMTS5, and PDGF-BB. The results are presented in Table 3.

### 3.4. Correlation Analysis between Plasma GAAGs and ADAMTS4, ADAMTS5 and PDGF-BB Levels in JIA Patients

Statistical analysis revealed a significant correlation between GAAGs levels and ADAMTS4 levels in blood obtained from patients before the start of ETA therapy (r = 0.846, *p* < 0.001) and in the third (r = 0.725, *p* < 0.001) month of its duration. On the other hand, analysis of correlations between GAAGs levels and ADAMTS5 levels in the blood of patients with JIA revealed significant correlations between the above-mentioned variables in patients in the 3rd (r = 0.482, *p* = 0.002) and 24th (r = −0.463, *p* = 0.003) month of biological therapy. There was also a significant relationship between GAAGs levels and PDGF-BB levels in the blood of patients before biological therapy (r = 0.687, *p* < 0.001) as well as in the 3rd (r = 0.791, *p* < 0.001) and 12th (r = 0.386, *p* = 0.017) month of its duration. In other cases, the correlations between the evaluated variables were not statistically significant (*p* > 0.05), as shown in Table 3.

### 3.5. Correlation Analysis between Plasma COMP and ADAMTS4, ADAMTS5 and PDGF-BB Levels in JIA Patients

Statistical analysis of correlations between plasma COMP levels and ADAMTS 4 revealed significant correlations between the above-mentioned variables in the 6th (r = 0.642, *p* < 0.001), 12th (r = 0.650, *p* < 0.001), 18th (r = 0.758, *p* < 0.001), and 24th (r = 0.770, *p* < 0.001) months of the therapy. On the other hand, analysis of correlations between COMP levels and ADAMTS5 levels in the blood of patients with JIA revealed significant correlations between the above-mentioned variables in patients before biological therapy (r = 0.348, *p* = 0.032), 6th (r = 0.646, *p* < 0.001), and 12th (r = 0.366, *p* = 0.024) month of therapy. There was also a significant correlation between COMP levels and PDGF-BB levels in blood obtained from patients at the 3rd (r = 0.381, *p* = 0.018) and 12th (r = 0.322, *p* = 0.049) month of therapy. In other cases, the correlations between the evaluated variables were not statistically significant (*p* > 0.05), as shown in Table 3.

### 3.6. Correlation Analysis between Plasma YKL-40 and ADAMTS4, ADAMTS5 and PDGF-BB Levels in JIA Patients

The analysis of the correlation between plasma YKL-40 concentration and ADAMTS4 levels revealed the presence of significant correlations between the concentrations of the above-mentioned parameters in 3rd (r = 0.336, *p* = 0.039), 12th (r = 0.360, *p* = 0.026), and 24th (r = 0.538, *p* < 0.001) month of ETA treatment. On the other hand, analysis of correlations between YKL-40 levels and ADAMTS5 levels in the blood of patients with JIA revealed significant correlations between the above-mentioned variables in patients in the 3rd (r = 0.506, *p* < 0.001), 12th (r = 0.574, *p* < 0.001), and 18th (r = 0.491, *p* = 0.002) month of biological therapy. Furthermore, there was a significant correlation between plasma YKL-40 levels and PDGF-BB levels in the patients in 3rd (r = 0.708, *p* < 0.001), 6th (r = 0.587, *p* < 0.001), 12th (r = 0.436, *p* = 0.006), and 24th (r = 0.656, *p* < 0.001) month of therapy. In other cases, the correlations between the evaluated variables were not statistically significant (*p* > 0.05), as shown in Table 3.

**Table 3 jcm-11-05069-t003:** Correlation analysis between plasma GAAGs, COMP, and YKL-40, and ADAMTS4, ADAMTS5, and PDGF-BB levels in JIA patients.

Parameter	Patients of JIA (n = 38)					
	Before ETA Treatment T0	Time after Starting Etanercept Therapy				
		3 Months T3	6 Months T6	12 Months T12	18 Months T18	24 Months T24
GAAGs						
ADAMTS4 r (p)	0.846 (*p* < 0.001)	0.725 (*p* < 0.001)	0.158 (NS)	0.211 (NS)	0.060 (NS)	−0.124 (NS)
ADAMTS5 r (p)	0.099 (NS)	0.482 (*p* = 0.002)	−0.045 (NS)	0.232 (NS)	0.155 (NS)	−0.463 (*p* = 0.003)
PDGF-BB r (p)	0.687 (*p* < 0.001)	0.791 (*p* < 0.001)	0.084 (NS)	0.386 (*p* = 0.017)	0.263 (NS)	0.167 (NS)
COMP						
ADAMTS4 r (p)	0.026 (NS)	0.179 (NS)	0.642 (*p* < 0.001)	0.650 (*p* < 0.001)	0.758 (*p* < 0.001)	0.770 (*p* < 0.001)
ADAMTS5 r (p)	0.348 (*p* = 0.032)	0.306 (NS)	0.646 (*p* < 0.001)	0.366 (*p* = 0.024)	−0.087 (NS)	0.446 (NS)
PDGF-BB r (p)	0.128 (NS)	0.381 (*p* = 0.018)	0.065 (NS)	0.322 (*p* = 0.049)	−0.029 (NS)	0.160 (NS)
YKL-40						
ADAMTS4 r (p)	0.113 (NS)	0.336 (*p* = 0.039)	−0.215 (NS)	0.360 (*p* = 0.026)	0.017 (NS)	0.538 (*p* < 0.001)
ADAMTS5 r (p)	0.016 (NS)	0.506 (*p* < 0.001)	−0.211 (NS)	0.574 (*p* < 0.001)	0.491 (*p* = 0.002)	0.066 (NS)
PDGF-BB r (p)	0.074 (NS)	0.708 (*p* < 0.001)	0.587 (*p* < 0.001)	0.436 (*p* = 0.006)	0.656 (*p* < 0.001)	−0.056 (NS)

Results are expressed as mean ± standard deviation; ETA, etanercept; JIA, Juvenile idiopathic arthritis; GAAGs, galactosaminoglycans; COMP, cartilage oligomeric matrix protein; YKL-40, human cartilage glyco-protein 39; PDGF-BB, plateled-derived growth factor BB; ADAMTS4, disintegrin and metal-loproteinase with thrombospondin motifs 4; ADAMTS5, disintegrin and metalloproteinase with thrombospondin motifs 5; NS, not statistically significant.

## 4. Discussion

In the course of JIA, proinflammatory compounds have been shown to promote structural and functional ECM cartilage changes. This occurs via multiple mechanisms involving modification of the activity of agents, which have catabolic or anabolic impact of matrix components. Pathological changes initially involve the articular surface, and as the disease progresses, lead to the loss of the characteristic layered architecture of cartilage tissue. Difficult-to-regenerate lesions involve the collagenous stroma are preceded by reversible changes in the proteoglycans network [20,21,22]. In our study, we confirmed the disorders of PGs metabolism, manifested by a significant reduction in plasma GAAGs level, in patients with an aggressive form of JIA, i.e., a disease requiring anti-cytokine therapy. However, the obtained results cannot be compared with those of other authors. This is due to the fact that plasma GAAGs in JIA patients treated with ETA have not been analyzed so far. It should be noted that glycans assessment was performed in children with JIA treated with methotrexate. The reduction in the total pool of glycosaminoglycans (GAGs), resulting from a decrease in the dominant glycan fraction, i.e., chondroitin sulfates, in blood [23,24], urine [23,24], and joint fluid [25] of JIA patients, was found. In contrast, plasma levels of dermatan sulfates, hyaluronic acid, and keratan sulfates increased, while heparan sulfates did not change in these patients. Based on these studies, it can be concluded that different types of GAGs respond differently to factors leading to the manifestation of pathological changes of the osteoarticular system in children with JIA [5,23,24,25,26]. For instance, excessive release of DS chains and HA chains of lower molecular mass seem to contribute to JIA onset. The listed GAGs—through modifying the body’s response to infectious agents including *Streptococcus pyogenes*, as well as by stimulating pro-inflammatory factors, i.e., IL-8, IL-12, TNF-α, or nitric oxide synthase—can lead to the manifestation of genetic predispositions underlying the development of the discussed disease. In contrast, the observed significantly lower plasma CS concentration in children with JIA, which correlates negatively with CRP, probably reflects “draining” of tissue pool of these GAGs, which results from the anti-inflammatory and anti-oxidative functions of CS. As a result of the above, the tissue metabolism of ECM components changes, which as a consequence manifests itself in changes in their concentration in the blood [5,23,25,27,28].

The significantly lower blood levels of GAAGs in patients qualified for ETA therapy due to the ineffectiveness or intolerance of the previous treatment with MTX, SSA, and EC are probably the result of the significantly increased degradation of ECM components occurring in the early JIA stages.

When the clinical symptoms of JIA are apparent, the tissue content of GAAGs is significantly reduced, and the synthesis processes of assessed compounds do not balance the magnitude of their degradation [5,20,23]. Therefore, the concentration of GAAGs in the blood of children with JIA before starting ETA treatment is significantly lower compared to the concentration of these compounds in the blood of healthy children. Among the factors involved in the degradation processes of ECM components, resulting in changes in the concentration of GAAGs in the blood are ADAMTS family aggrecanases and matrix metalloproteinases [3,5]. In the study, we demonstrated a significant correlation between the levels of GAAGs and ADAMTS4 in blood obtained from patients before ETA therapy and after 3 months of therapy. In view of these results, it can be assumed that ADAMTS4 has a leading role in modeling particularly pathological transformations of ECM components, whereas ADAMTS5 seems to performs the main function in osteochondrial development and aggrecan metabolism occurring under physiological conditions [29,30,31]. The role of aggrecanases in the remodeling of ECM in the course of JIA is described in detail in our earlier work [11]. In addition, we showed that the two-year use of anti-cytokine therapy in patients resulted in clinical improvement in patients but did not normalize GAAGs levels.

Joint dysfunction in patients with JIA, preceded by proteolysis of matrix components, may also be promoted by impaired biosynthesis of these components. Numerous anabolic growth factors, such as TGF-β, IGF-1, or PDGF-BB, are capable of stimulating various cellular processes, i.e., proliferation, migration, or morphogenesis during tissue development and healing. The aforementioned factors bind to ECM proteins or heparan sulfates. Hence, it is believed that the ECM, in addition to storing and protecting these factors from degradation, plays a major role in controlling their signaling. These factors in return stimulate the synthesis of ECM components and its remodeling. Changes in the activity of the discussed factors are revealed in patients with osteoarticular disorders [32,33]. This thesis is confirmed by the results of study performed in the present work. Significantly higher plasma PDGF-BB concentrations were found in children with JIA who qualified for etanercept treatment. Moreover, it was demonstrated that the two-year application of anti-cytokine therapy, which contributed to clinical improvement in patients, simultaneously led to normalization of the levels of this marker. As shown in other studies, high PDGF-BB levels also characterize patients with a newly established diagnosis of JIA even before disease-modifying treatment is applied to patients, as well as patients with limited juvenile systemic scleroderma or patients with systemic lupus erythematosus [20,34,35].

The association of PDGF-BB with GAAGs in patients with active forms of JIA, found by us, confirms a significant preventive and regenerative role of this compound in relation to the damage of ECM structures, which are caused by pro-inflammatory influences. The former as an anabolic factor acts as a mitogenic and chemotactic agent for cells of mesenchymal origin, including cartilage tissue cells. Moreover, receptors for PDGF were found on many cell types, including chondrocytes, and their amount increases in the presence of proinflammatory cytokines such as IL-1 [36,37].

PDGF, in addition to stimulating the chondrocyte proliferation and chondrocyte PGs biosynthesis, is also a stimulator of fibroblast-like synoviocytes (FLSs). The latter are cells that secrete many ECM components such as collagens, fibronectin, laminin, and proteoglycans. Moreover, activated FLSs also become a source of proteases, i.e., enzymes capable of depolymerizing proteoglycan components of osteoarticular structures [36]. However, PDGF has been found to inhibit collagenase synthesis while increasing IL-1β-induced synthesis of proinflammatory prostaglandin E2 by FLSs. Hence, it can be concluded that, stimulated by PDGF, FLSs play an important role in the processes of both PGs/GAGs synthesis and degradation. It has been shown that PDGF strongly and selectively enhances cytokine-induced synthesis and secretion of proinflammatory factors by FLSs, such as IL-6, IL-8, macrophage inflammatory protein 1α, and metalloproteinases 3 (MMP-3) [37]. Increased activity of matrix metalloproteinases has been reported in children with JIA, resulting in the pattern of extracellular matrix remodeling markers of articular structures observed in the blood samples from children with active JIA [38,39].

The changes in plasma concentrations of GAAGs in JIA patients qualified for ETA treatment was accompanied by decrease of circulating COMP. This confirms the significant structural abnormalities of the cartilage matrix in children with JIA. This is due to the fact that predominant tissue localization of COMP is articular cartilage, while small amounts of the protein in question are also found in tendons, meniscus, or synovial membrane [40]. Deficiency of this protein may, on the one hand, promote the degradation of collagenous cartilage tissue, but on the other hand, COMP deficiency may result from pathological joint remodeling of the ECM. Hence, COMP is considered by researchers as a potential marker of cartilage degradation, as well as an indicator of disease progression and treatment effects [41,42]. We showed that the two-year anticytokine therapy, which leads to the extinction of the inflammatory process and clinical improvement, simultaneously contributes to the normalization of plasma levels of the assesed marker in patients undergoing therapy. Thus, our results and those of Bjørnhart et al. [43] and Urakami et al. [44] confirm the above thesis. In contrast to our results, Struglics et al. [40] showed that JIA patients are characterized by an increase in blood COMP levels. These discrepancies are likely the result of the cited authors’ qualification of patients into three types of arthropathy, different disease duration, and inconsistent pharmacotherapy, which might determine the turnover of the ECM, since we have shown that COMP metabolism in patients with JIA, both before and during ETA therapy, remains related to the activity of proteolytic processes, stimulated by ADAMTS4 and ADAMTS5 concentrations, as well as anabolic processes, related to PDGF-BB concentrations. A similar relationship, albeit with other enzymes of the family in question, was found in a study by Liu et al. [42]. The cited authors showed that ADAMTS group enzymes, namely ADAMTS7 and ADAMTS12, are overexpressed in cartilage and synovial membrane in osteoarthritis patients, contributing to COMP degradation. Moreover, antibodies against ADAMTS7 and/or ADAMTS12 have been shown to inhibit COMP degradation in vitro, induced by the effects of TNF-α and IL-1β. Moreover, the researchers found that silencing with small interfering RNA in human chondrocytes leads to inhibition of ADAMTS7 or ADAMTS12 expression and significantly prevents COMP degradation. Accordingly, the cited authors indicate that the ADAMTS7 and ADAMTS12 enzymes are newly identified proteinases responsible for the COMP degradation observed in joint inflammation [42].

A component of the cartilaginous matrix, the metabolism of which is associated with the inflammatory process, is, as also determined in this study, YKL-40. Increased expression of YKL-40 mRNA has been described in a subpopulation of macrophages involved in inflammatory processes and matrix remodeling in cartilage tissue [7]. In our study, we found significantly higher levels of YKL-40 in the blood of children with untreated JIA compared to the controls. In healthy subjects, low blood levels of YKL-40 are found. Thus, high concentrations of YKL-40 in biologically untreated children with JIA confirm intra-articular synthesis of the glycoprotein by synovial cells resembling fibroblasts, by neutrophils or macrophages and chondrocytes [45,46]. The influence of high blood concentrations of factors stimulating anabolic processes of ECM components on the synthesis of the glycoprotein in question is also not excluded. The significant correlations shown in the blood of JIA patients between PDGF-BB and YKL-40 concentrations seem to confirm the above mechanism. Although the metabolic pathways described above have not been confirmed in children with JIA, their presence is highly probable. High concentrations of YKL-40 in the blood of children with aggressive arthropathy indicate that its metabolism is related to the activity of the inflammatory process. This thesis is supported by the normalization of YKL-40 concentrations in the blood of patients with clinically compensated JIA, i.e., after two years of ETA therapy. Other studies [47,48,49] have shown that in patients with active RA, the initially observed high YKL-40 concentrations were reduced after effective therapy. In contrast, in patients whose rheumatic disease remained clinically active despite treatment, further increases in serum YKL-40 levels were observed [49]. The above-mentioned studies have proven that the concentration of YKL-40 in blood reflects metabolic changes in cartilage, while being an expression of the local, i.e., cartilaginous, course of the inflammatory process [46,49]. The involvement of YKL-40 in the local transformation of cartilage ECM components is confirmed by its significant associations with ADAMTS4 and ADAMTS5, which are specific for aggrecan metabolism. Although the associations between these variables have not yet been confirmed in other studies, YKL-40 is known to bind to proteoglycans and collagens and affect their biosynthesis [50].

Although the results of this study are significant, there are some limitations that affect the generalizability of the findings. First, the unsatisfactory sample size, due to COVID-19-dependent difficulties in obtaining blood samples within the planned timeframe, means that further studies with a large sample are needed to evaluate the utility of GAAG, COMP, and YKL-40 as biomarkers of disease progression and to assess the effectiveness of treatment. Second, special care should be taken in generalizing these results because this study was conducted on patients only with oligo- and polyarticular types of JIA. Further studies are desirable to investigate changes in the parameters assessed in other types of JIA. Thirdly, the assessment of joint markers was performed in the patients’ blood, due to the fact that this is a readily available material, while its acquisition is one of the minimally invasive procedures, unlike joint synovium. In turn, the planned experiment required the material to be collected 6 times in successive months of therapy.

## 5. Conclusions

To summarize, in the course of JIA, the metabolism of ECM components of articular cartilage is regulated by different mechanisms, depending both on the hyperactivity of depolymerizing factors, including ADAMTS4 and ADAMTS5, and the influence of anabolic compounds, including PDGF-BB. The observed concentration profile of the assessed ECM remodeling markers, i.e., GAAGs, COMP, and YKL-40, in biologically treated children with JIA seems to indicate that COMP and YKL-40 may be useful biomarkers of the disease course, assessment of treatment efficacy, and long-term periodic follow-up after completion of therapy for early detection of possible arthropathy recurrence. On the other hand, the profile of changes in GAAGs concentrations in the blood of patients with arthropathy seems to reflect systemic changes in the extracellular matrix of connective tissue. The results of our study provide a rationale for expanding the existing panel of biochemical tests useful in diagnosing and monitoring the treatment of juvenile idiopathic arthritis. The understanding of changes in ECM components of connective tissue in the course of this pediatric arthropathy may allow the introduction of both new diagnostic tools and new therapeutic strategies in children with JIA.

## Figures and Tables

**Table 2 jcm-11-05069-t002:** The distribution patterns of plasma GAAGs, COMP, YKL-40, and PDGF-BB in the healthy individuals (control subjects) and JIA patients.

Parameter	Control Subjects (n = 30)	Patients of JIA (n = 38)					
		Before ETA Treatment T0	Time after Starting Etanercept Therapy				
			3 Months T3	6 Months T6	12 Months T12	18 Months T18	24 Months T24
GAAGs (ug/mL)	18.98 ± 4.91	9.37 ± 2.10 ^a^	21.98 ± 6.22 ^b^	23.23 ± 5.82 ^b^	18.71 ± 4.45 ^b,c,d^	15.18 ± 3.85 ^b,c,d,e^	16.13 ± 2.85 ^a,b,c,d^
COMP (ng/mL)	597.25 ± 100.89	467.18 ± 116.26 ^a^	604.95 ± 152.03 ^b^	611.40 ± 135.64 ^b^	640.52 ± 176.28 ^b^	625.75 ± 110.44 ^b^	603.99 ± 115.20 ^b^
YKL-40 (ng/dL)	39.66 ± 12.73	78.29 ± 27.62 ^a^	51.14 ± 18.77 ^b^	45.98 ± 17.46 ^b^	43.10 ± 16.86 ^b^	39.57 ± 15.15 ^b^	34.38 ± 13.14 ^b,c^
PDGF-BB (ng/mL)	0.87 ± 0.36	1.31 ± 0.38 ^a^	1.12 ± 0.41	0.99 ± 0.33 ^b^	0.90 ± 0.29 ^b^	0.95 ± 0.34 ^b^	0.87 ± 0.33 ^b,c^

Results are expressed as mean ± standard deviation; ETA, etanercept; JIA, Juvenile idiopathic arthritis; GAAGs, galactosaminoglycans; COMP, cartilage oligomeric matrix protein; YKL-40, human cartilage glycoprotein 39; PDGF-BB, plateled-derived growth factor BB; ^a^
*p* < 0.05 compared to control group; ^b^
*p* < 0.05 compared to T0 group; ^c^
*p* < 0.05 compared to T3 group; ^d^
*p* < 0.05 compared to T6 group; ^e^
*p* < 0.05 compared to T12 group.

## Data Availability

The datasets analyzed or generated during the study are available from the author: klaudia_092@vp.pl.

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
