# Peer review of "GAAGs, COMP, and YKL-40 as Potential Markers of Cartilage Turnover in Blood of Children with Juvenile Idiopathic Arthritis Treated with Etanercept—Relationship with ADAMTS4, ADAMTS5, and PDGF-BB"

_jcm, 2022, doi:10.3390/jcm11175069_

Round 1
Reviewer 1 Report
The main weak point of the manuscript: small sample size, assessment of articular-related markers in blood not in synovial fluid, unbalanced groups for assessment makes the results of the study doubtful. Manuscript required Limitations section. The study results are difficult to evaluate and many other factors influenced study results.
Reviewer 2 Report
This is an interesting manuscript focused on potential biomarkers of joint dysfunction and indicators of efficacy of biologic therapy, especially in children with juvenile idiopathic arthritis treated with etanercept. They quantified galactosaminoglycans (GAAGs), oligomeric cartilage matrix protein (COMP) and human cartilage glycoprotein (YKL-40) in blood, which can be used to assessing the chondroprotective biological efficacy of therapy (COMP and YKL-40) and reflect systemic extracellular matrix transformations during JIA (GAAGs). The study is well-conducted, the methods are correct and the results of provide those potential biomarkers of biochemical tests useful in diagnosing.
However, some details need to be refined in this manuscript.
1. In the introduction, the authors introduced COMP and YKL-40 as biomarkers, but why these two were chosen and how they were selected from so many potential proteins are important for the logic of this manuscription.
2. This study confirmed the disorders of PGs metabolism, manifested by a significant reduction in plasma GAAGs level, which cannot be compared with other authors results. Please refine the discussion on whether there are other possibilities that lead to different results from other authors.
3. At the end of discussion, YKL-40 is first confirmed by its significant associations with ADAMTS4 and ADAMTS5, which are specific for aggrecan metabolism. The results are interesting and innovative, if more follow-up experiments will be conducted, which will be more convincing.
These three markers (GAAGs, COMP and YKL-40) significantly speed up the biochemical test’s efficiency after treatment of juvenile idiopathic arthritis. This method is of great value in both new diagnostic tools and new therapeutic strategies in children with JIA. It would be accepted if the authors can supplement and modify the manuscript well.
Round 2
Reviewer 1 Report
Dear Authors!
The manuscript became better and might be accepted in this stage.